**Data Availability Statement:** The data files generated and analyzed in this study are publicly available from the Data Archiving and Networked

# The effect of preferred music versus disliked music on pain thresholds in healthy volunteers. An observational study

Hans Timmerman[1¤a]*, Regina L. M. van Boekel[1], Ludo S. van de Linde[1¤b], Ewald M. Bronkhorst[2], Kris C. P. Vissers[1], Selina E. I. van der Wal[1], Monique A. H. Steegers[1¤c]

1 Department of Anesthesiology, Pain and Palliative Medicine, Radboud University Medical Center, Nijmegen, The Netherlands, 2 Department for Health Evidence, Radboud University Medical Center, Nijmegen, The Netherlands

¤a Current address: Department of Anesthesiology, Pain Center, University of Groningen, University Medical Center Groningen, Groningen, the Netherlands
¤b Current address: Stichting Pro Persona Ggz, Wolfheze, the Netherlands
¤c Current address: Department of Anesthesiology, Amsterdam University Medical Center, Location VU, Amsterdam, the Netherlands
* hans.timmerman@radboudumc.nl, h.timmerman02@umcg.nl

## Abstract

Pain is a prevalent and debilitating healthcare problem. Since pharmacological treatments have numerous side-effects, additional treatment could be beneficial. Music has been shown to affect the pain perception and the pain threshold. The objective of this observational study was to evaluate the effect of preferred music as opposed to disliked music on pain (tolerance) thresholds and perceived pain intensity in healthy volunteers. Pain thresholds were measured via quantitative sensory testing. The volunteers were randomly assigned to either handheld pressure algometry to assess the pressure pain threshold to or electrical measurements to assess the electrical pain tolerance threshold while listening to preferred and disliked music. The pain thresholds were administered on the dorsal side of the forearm. The perceived pain intensity was assessed via a numerical rating scale, ranging from 0 (no pain) to 10 (worst pain imaginable). In total 415 volunteers were included in this study. The pressure pain threshold was assessed in 277 volunteers and in the electrical pain tolerance threshold test 138 volunteers were entered. In both groups, preferred music yielded higher pain thresholds than disliked music (*P*<0.001) and lower perceived pain intensity during the stimulus (P = 0.003). Moreover, the highest pain thresholds of both pressure pain and electrical pain tolerance thresholds were obtained when the preferred music was preceded by disliked music. Listening to preferred music when receiving noxious stimuli leads to higher pain thresholds and lower perceived pain scores in comparison with disliked music. Preferred music could be beneficial for patients with pain or undergoing painful procedures.

Services (DANS) database (https://doi.org/10.17026/dans-zsg-4anh).

**Funding:** The author(s) received no specific funding for this work.

**Competing interests:** The authors have declared that no competing interests exist.

## Introduction

Pain is defined as "an unpleasant sensory and emotional experience associated with, or resembling that associated with, actual or potential tissue damage" [1]. Acute pain is a common sensation in daily life and serves as a warning system for the body. The pain is caused, by example, by injury, disease or a medical procedure. In general, it lasts for only a short period (up to three months) and normally disappears when the cause of the pain is treated. Chronic pain is seen as a result of persistent maladaptive pain. It is called chronic if it surpasses the normal tissue healing time and no longer serves a physiological purpose. Most often a duration of pain of more than three months is used [2]. The transition of acute pain into chronic pain is an observed phenomenon and described in many clinical settings [3,4]. Reducing the risk of acute pain becoming chronic will lead to more quality of life of the patient and a lower impact on society in general [5]. Acute as well as chronic pain is associated with a decline in physical, mental, social and spiritual health and quality of life, rising medical costs and in patients with chronic pain associated with an increase in demand and dependency of prescription opioids [6,7].

As drugs have a range of negative side effects, additional non pharmacological pain treatments can be beneficial. It is showed that music interventions may provide an effective approach in the treatment of acute (postoperative) pain as well as in chronic pain [8]. Music could serve as an inexpensive therapeutic option complementary to conventional pain treatments such as pharmacological interventions. It is easily accessible and can be self-administered by patients, thereby enhancing their feeling of empowerment [9]. Recent studies have supported music's beneficial effects on pain perception. The analgesic effect of music was showed in research carried out on patients with burns who, when music was played during the routine wound care, experienced less pain compared with the patient control group [10]. According to a recent systematic review including a meta-analysis [11], music reduced pain intensity but also resulted in a lower level of anxiety, less use of analgesics and listening to music increased patients' satisfaction in the postoperative phase. Moreover, the authors concluded that a patient should preferably listen to music of their own choice. In a Cochrane systematic review by Bradt and colleagues [12] a large pain reducing effect of music interventions was noted in adding music to the care in patients with cancer compared to standard care. In a systematic review about music-induced analgesia in patients with chronic pain [13] it was found that music reduced self-reported chronic pain intensity and depressive symptoms most effectively when the patients could make their own choice for the applied music. In a study by Hekmat and Hertal [14] it was showed that preferred music increased the pain tolerance and lowered self-reported pain intensity during an ice-water hand immersion test compared to the same task without music. However, in the same study, in groups were participants listened to disliked music compared to no music the participants showed no differences in pain tolerance and self-reported pain. When listening to preferred music compared to relaxation music painful stimuli were tolerated for a longer period, but only in females the experienced pain intensity was lower [15]. Paller and colleagues concluded in their review [16] that women in comparison to men reported more pain in a clinical situation but also reported a higher sensitivity to pain in an experimental setting.

Until now it is not completely clear how music influences pain [17,18]. One of the possible mechanisms of how music influences pain processing is distraction, a process wherein music binds cognitive abilities, distracts the participants' attention and subsequently inhibits the intensity and unpleasantness of pain [19]. Having an emotional connection with the music when listening also might play a role in the pain reducing abilities based on the stimulation of positive emotions and pain modulation. Moreover, music might also have an effect on the activity of the autonomic nervous system [20].

Our hypothesis is that listening to preferred music would yield higher pain thresholds, whereas listening to disliked music would result in lower thresholds. Therefore, the aim of this research is to evaluate the effect of preferred music as opposed to disliked music on the perception of pain and the pain thresholds in healthy volunteers.

## Materials and methods

### Study design and setting

This observational study was conducted in 2016 during "Lowlands", a festival mainly focussing on music, but also with attention to, by example, literature, ballet and science. This festival is located in Biddinghuizen (the Netherlands) and the organization provided a designated venue for science and research. The Committee on Research Involving Human Subjects region Arnhem-Nijmegen ethically reviewed the study protocol in line with the Dutch law before the conduction of the study (file number CMO: 2016–2784).

### Participants

Recruitment of healthy volunteers lasted throughout the three days of the festival. Interested volunteers were asked to participate by the researchers of this study when they were passing by the research venue. When they were (1) at least 18 years old, (2) in normal health (no reported diseases and/or illness) and (3) had sufficient knowledge of the Dutch language they were able to participate in this study. Volunteers received no financial compensation or gifts for their participation. Exclusion criteria for participation were (1) the use of analgesic drugs in the preceding twelve hours, (2) the use of antidepressants, (3) pain in the arm, neck or shoulder (uni- or bilaterally), (4) cardiac diseases, (5) psychiatric or neurological diseases, (6) injury to the forearms or hands, (7) Raynaud disease, (8) pregnancy, (9) a blood alcohol content (BAC) of $> 220$ μg/l or (10) the use of recreational drugs in the past 24 hours. A written informed consent was obtained from every participant, after which the BAC was determined by a physician via a digital breath alcohol analyzer (ALCOSCAN AL9000 Lite, Sentech, Gyeonggi-Do, Republic of Korea).

### Measurements

**Demographics.** Sex, age, height, weight and left- or right hand dominance were recorded.

**Electrical Pain Tolerance Threshold (EPTT).** The EPTT was defined as the maximum level of pain which the participant could tolerate. The EPTT was expressed in milli- ampere (mA). The QST-3 device (JNI Bio-medical ApS, Klarup, Denmark) was used to apply the electrical current onto volunteers' skin via disposable electrodes (Kendall ECG Electrodes, H34SG, $50 \times 45$ mm; Covidien, Mansfield, MA, USA). The electrical current was set as a tetanic stimulation (100Hz, 0.2 ms square waves) with a ramping rate of 1 mA per second. The current started at 0 mA and the maximum electrical current was limited to 50mA because of safety regulations. Volunteers were instructed beforehand, in advance to start the electrical current by pressing the button and to let go of the button as soon as the pain became intolerable. EPTT was assessed four times including a training session, a baseline measurement in silence and the experimental measurements during preferred and disliked music.

**Pressure Pain Thresholds (PPT).** To assess the PPT the digital pressure algometer (Wagner Instruments, Force TEN™ Digital Force Gage FDX 50, Greenwich, CT, USA) with a 1.0cm$^2$ probe was placed under a 90˚ angle on the forearm muscles of the non-dominant arm. PPT was expressed in N/cm$^2$ and applied with a ramping rate of $\sim 5$ N per second controlled via visual feedback on the algometer display. Pressure started at 0 N and was limited to 220 N/

$cm^2$ for safety reasons. The volunteers were instructed to raise their hand and/or to say "stop" when they felt a burning, painful, or stinging sensation alongside the pressure. PPT was assessed four times, including a training session, a baseline measurement in silence and the experimental measurements during preferred and disliked music.

**Perceived pain intensity.** Volunteers were asked to rate their perceived pain intensity directly after the assessment of the PPT or the EPTT on a numerical rating scale (NRS). The NRS is a 11-point Likert scale ranging from zero (0, no pain) to 10 (worst imaginable pain). The psychometric properties of the NRS are considered to be good [21].

## Measurement locations

The measurements (EPTT and PPT) were performed on the dorsal side of the non-dominant forearm. This location was selected because there is muscle volume to assess the PPT and it was simply approachable by uncovering the forearm. The measurement locations were marked in advance on the patient's skin by using a ruler and a marker. For the PPT, the locations were located at 6 (M0), 8 (M1), 10 (M2) and 12 (M3) centimetres from the styloid process of the ulna. The training session was performed at the M0 location, baseline measurement in silence at M1 and the experimental stimuli were given at M2 and M3. For the EPTT, all measurements took place at M0 and M2.

## Study procedures

The measurements obtained in this study were based on the methodology of the Nijmegen Aalborg Screening Quantitative Sensory Testing (QST), also called NASQ protocol [22–25]. The measurements during the festival were performed by four physicians, four researchers and four students. The methodology of the study and the collection of the data of our volunteers has been described earlier [26].

During the preparation of the study, the twelve male and female operators were meticulously trained during two training sessions in the approach and instruction of the volunteers and in the uniformly conduction of both the tests stimuli and the assessment of the perceived pain.

The random assignment of the volunteers took place by letting volunteers pick an opaque, sealed envelope. They were assigned to the electrical pain tolerance threshold group (EPTT) or the pressure pain threshold (PPT) in a 1:2 ratio (because of the availability of the equipment in the measurement stations), whilst at the same time being counterbalanced between music protocol A or B (1:1 ratio). In protocol A, volunteers listened to their preferred music first, before listening to their disliked music. In protocol B, this was the other way around (disliked music first, preferred music second). We used these two protocols to avoid an order effect of the music presented. Volunteers were aware of which experimental and protocol group they were assigned to.

The instructions for the volunteers were standardized and read from paper by the operators. To avoid distraction and maximize attention the volunteers were seated on a chair at tables in identical cubicles opposite to the operator who was performing the tests. In two cubicles EPTT was measured, and in four cubicles PPT was assessed. The volunteers were asked to state their preferred song and their most disliked song before commencement of the measurements. All music was available via Spotify (Spotify, Stockholm, Sweden). The music was provided through earphones during the experimental measurements (in-ear plugs, JBL C100si, Los Angeles, USA) by the research staff. During the measurements volunteers wore protective industrial earmuffs (3M Peltor Optime III H540, Maplewood, MN, USA) over the in-ear earphones to minimize background sound. In order to get volunteers acquainted with the stimuli,

measurements started with a training measurement with pressure or electrical pain stimuli. Hereafter the pressure or electrical pain stimuli were administered in a relatively silent setting in order to acquire a baseline measurement. Subsequently, volunteers listened to 60 seconds of their preferred or their disliked music. Afterwards they received pressure or electrical pain stimuli, while they continued listening to their music. When they released the button of the EPTT or in case of the PPT signalled (by putting up their other hand) their maximum pain level was reached the painful stimuli stopped immediately. After this, the music was stopped. Volunteers were asked to rate their perceived pain intensity by the NRS during all measurements. The music started again with either preferred or disliked music depending on the study protocol (protocol A or B) and the procedure and measurements were repeated.

## Statistical methods

An estimation of the sample size was based on a previously carried out pilot study (not published). This pilot study revealed a spread of 20 N in PPT outcomes. A difference of 5 N was considered to be statistically significant. In order to obtain a power of 90% and a significance level of 5%, a sample size of 170 volunteers was needed for the PPT group. Since no similar data exists for the EPTT, the same number of volunteers was deemed necessary for this group. As a result, a minimum of 340 volunteers was needed. Data entry was performed using Castor Electronic Data Capture (CIWIT b.v., Amsterdam, the Netherlands).

The effect of music was estimated by multilevel regression models with "type of music" (preferred music versus disliked music) and "timing" (first or second exposition to music) as independent variables. As each participant was measured twice, all regression models were supplied with a random intercept. These analyses were performed independently for both physical stimuli. For both stimuli both the maximum pain level (NRS score) as well as the maximum level of stimulation (either mA or N) were analyzed. The statistical analysis was carried out using R version 3.6.2, and the LME4 library (version 1.1–21) [27]. $P \leq 0.05$ was considered statistically significant for all tests. The data files generated and analyzed in this study are available from Data Archiving and Networked Services (DANS): https://doi.org/10.17026/dans-zsg-4anh [28].

## Results

### Characteristics of the volunteers

The enrolment process is shown in Fig 1. In total 484 volunteers were assessed for participation and 417 of them were included in the study. 67 volunteers were excluded, because of not fulfilling the inclusion criteria (n = 1), refusal to participate (n = 8), alcohol level above > 220 μg/l (n = 39), consumed recreational drugs (n = 10), suffering from a painful arm (n = 1) or an open wound on the arm (n = 1), having neurological (n = 1), psychiatric (n = 1) or cardiac diseases (n = 1), or used antidepressant (n = 2) or analgesic drugs (n = 2).

The EPTT group was further randomly assigned into a subgroup receiving their music according to protocol A (n = 69) and B (n = 69). This was also done for the PPT group, wherein 140 volunteers were randomly assigned to protocol A and 139 volunteers to protocol B. During data entry the data of two volunteers in the PPT group were excluded from the study as it was unclear which protocol had been followed, so the EPTT group and PPT group consisted of 138 and 279 volunteers respectively. The data of these remaining 415 subjects was used for the data-analysis.

The baseline demographical data is summarised in Table 1. The averages (± standard deviation(SD)) of age, height, weight and body mass index (BMI) in the EPTT group were respectively 27.8 (± 8.1) years old, 177 (± 8.6) cm, 74.8(± 13.9) kg and 23.7 (± 3.6) kg/m$^2$, whereas in

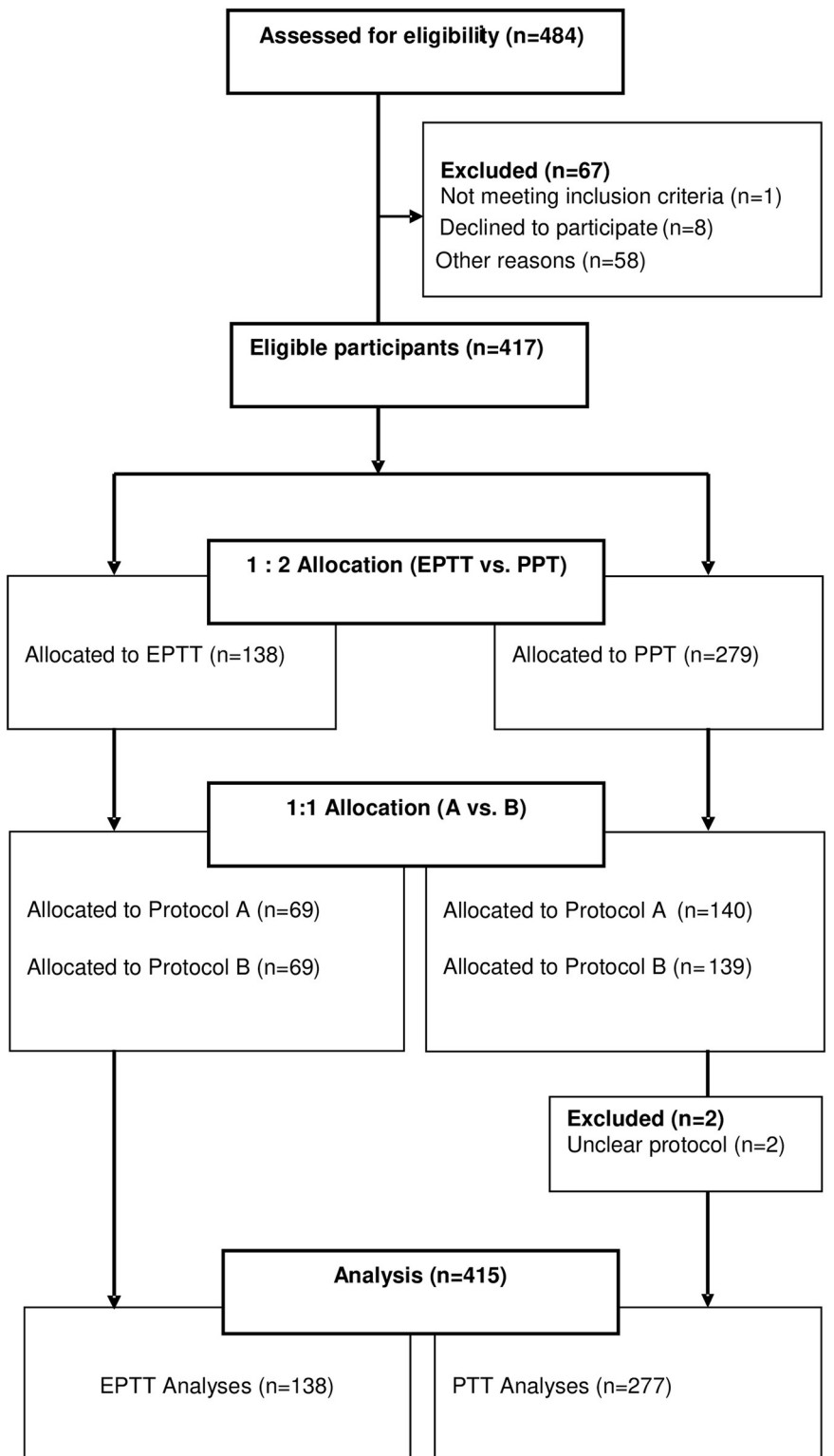

**Fig 1. Flow diagram of the study.** EPTT: Electrical Pain Tolerance Threshold; PPT: Pressure Pain Threshold; n: number; A: protocol A (preferred music first, disliked music second)); B: Protocol B (disliked music first, preferred music second).

**Table 1. Demographic characteristics at baseline.**

|  | Total group | EPTT | PPT |
|---|---|---|---|
| Number | 415 | 138 | 277 |
| Sex |  |  |  |
| *Male (%)* | 185 (44.6%) | 69 (50.0%) | 116 (41.9%) |
| *Female (%)* | 222 (53.5%) | 66 (47.8%) | 156 (56.3%) |
| *Missing n (%)* | 8 (1.9%) | 3 (2.2%) | 5 (1.8%) |
| Age (year)(mean ± SD) | 28.2 ± 8.6 | 27.8 ± 8.1 | 28.4 ± 8.9 |
| Height (cm)(mean ± SD) | 177 ± 9.2 | 177 ± 8.6 | 177 ± 9.4 |
| Weight (kg)(mean ± SD) | 74.1 ± 15.0 | 74.8 ± 13.9 | 73.8 ± 15.5 |
| BMI (kg/m$^2$)(mean ± SD) | 23.5 ± 3.8 | 23.7 ± 3.6 | 23.3 ± 3.8 |
| Random assignment |  |  |  |
| *Protocol A (preferred music first) n (%)* | 208 (50.1%) | 69 (50.0%) | 139 (50.2%) |
| *Protocol B(disliked music first) n (%)* | 207 (49.9%) | 69 (50.0%) | 138 (49.8%) |
| Left- or right-hand dominance |  |  |  |
| *Left-hand n (%)* | 57 (13.7%) | 21 (15.2%) | 36 (13.0%) |
| *Right-hand n (%)* | 351 (84.6%)) | 114 (82.6%) | 237 (85.6%) |
| *Missing n (%)* | 7 (1.7%) | 3 (2.2%) | 4 (1.4%) |
| Smoking |  |  |  |
| *Yes n (%)* | 73 (17.6%) | 27 (19.6%) | 46 (16.6%) |
| *No n (%)* | 329 (79.3%) | 108 (78.3%) | 221 (79.8%) |
| *Missing n (%)* | 13 (3.1%) | 3 (2.2%) | 10 (3.6%) |
| Medication |  |  |  |
| *Yes n (%)* | 62 (14.9%) | 25 (18.1%) | 37 (13.4%) |
| *No n (%)* | 350 (84.3%) | 112 (81.2%) | 238 (85.9%) |
| *Missing n (%)* | 3 (0.7%) | 1 (0.7%) | 2 (0.7%) |

EPTT: Electrical Pain Tolerance Threshold; PPT: Pressure Pain Threshold; SD: Standard Deviation; cm: Centimeter; kg: Kilogram; BMI: Body Mass Index; m2: Square metre; n: Number. Electrical and pressure pain thresholds and NRS scores per protocol in EPTT and PPT groups.

the PPT group this was 28.4 (± 8.9) years old, 177 (± 9.4) cm, 73.8 (± 15.5) kg and 23.3(± 3.6) kg/m$^2$.

In Table 2 the EPTT, PPT and NRS scores per protocol are shown. Per group we found the highest EPTT and PPT and the lowest perceived pain during listening to preferred music compared to disliked music as well as silence.

## The effect of preferred and disliked music on pain thresholds

In Table 3, the effect (β) of preferred and disliked music on pain thresholds and the sequence effect is shown. In order to calculate the effect of music on pressure and electrical pain thresholds, the pain thresholds in disliked music were subtracted from pain thresholds in preferred music. In both the EPTT and PPT groups, music exerted a statistically significant difference in pain thresholds in favour of preferred music (3.625 mA; $p < 0.001$ and 8.582 N; $p < 0.001$, respectively). In addition, a statistically significant, albeit small, difference in subjective pain levels (NRS) was found in favour of preferred music in both the EPTT and PPT groups (NRS: -0.261; $p = 0.003$ and NRS: -0.181; $p = 0.003$, respectively). The sequence effect was calculated by subtracting pain thresholds of the second measurement (preferred or disliked music, M2) from pain thresholds of the third measurement (disliked or preferred music, M3). Only in the EPTT group a statistically significant sequence effect was encountered (1.308 mA; $p = 0.003$).

**Table 2. Electrical and pressure pain thresholds and NRS scores per protocol in EPTT and PPT groups.**

| EPTT group | Protocol A (preferred music first) | | | Protocol B (disliked music first) | | |
|---|---|---|---|---|---|---|
| | M1 | M2 | M3 | M1 | M2 | M3 |
| | Silence | Preferred | Disliked | Silence | Disliked | Preferred |
| mA (mean ± SD) | 21.5 ± 10.2 | 23.8 ± 10.8 | 21.5 ± 10.6 | 25.3 ± 13.2 | 23.8 ± 12.1 | 28.8 ± 14.4 |
| NRS (mean ± SD) | 6.5 ± 2.2 | 6.4 ± 2.4 | 6.7 ± 2.2 | 7.1 ± 1.6 | 7.2 ± 1.6 | 7.0 ± 1.9 |
| PPT group | Protocol A (preferred music first) | | | Protocol B (disliked music first) | | |
| | M1 | M2 | M3 | M1 | M2 | M3 |
| | Silence | Preferred | Disliked | Silence | Disliked | Preferred |
| N/cm$^2$ (mean ± SD) | 66.4 ± 38.7 | 71.0 ± 39.5 | 60.7 ± 35.1 | 75.9 ± 43.3 | 70.3 ± 41.0 | 77.2 ± 42.2 |
| NRS (mean ± SD) | 3.6 ± 2.1 | 3.6 ± 2.3 | 3.6 ± 2.3 | 3.7 ± 2.2 | 3.9 ± 2.3 | 3.6 ± 2.3 |

EPTT: Electrical Pain Tolerance Threshold; PPT: Pressure Pain Threshold; SD: Standard Deviation; M1: Measurement 1; M2: Measurement 2; M3: Measurement 3; NRS: Numerical Rating Scale (perceived pain intensity, range 0–10); mA: Milli-Ampère; N/cm2: Newton (force) per cm2.

**Table 3. The effect of preferred and disliked music on pain thresholds and the sequence effect in the administered protocols.**

| EPTT group | | | | |
|---|---|---|---|---|
| **mA** | | | | |
| | **Effect (β)** | **CI-lower** | **CI-upper** | **P** |
| Intercept[#] | 20.690 | 18.305 | 23.076 | 0.000 |
| Protocol B vs A (0 = first; 1 = second) | 1.308 | 0.472 | 2.144 | 0.002 |
| Music Preferred vs Disliked | 3.625 | 2.789 | 4.462 | <0.001 |
| **NRS** | | | | |
| | **Effect (β)** | **CI-lower** | **CI-upper** | **P** |
| Intercept[#] | 6.891 | 6.464 | 7.319 | 0.000 |
| Protocol B vs A (0 = first; 1 = second) | 0.043 | -0.125 | 0.212 | 0.611 |
| Music Preferred vs Disliked | -0.261 | -0.429 | -0.093 | <0.001 |
| **PPT group** | | | | |
| **N/cm$^2$** | | | | |
| | **Effect (β)** | **CI-lower** | **CI-upper** | **P** |
| Intercept[#] | 67.964 | 62.526 | 73.403 | <0.001 |
| Protocol B vs A (0 = first; 1 = second) | -1.656 | -3.511 | 0.199 | 0.080 |
| Music Preferred vs Disliked | 8.582 | 6.727 | 10.437 | <0.001 |
| **NRS** | | | | |
| | **Effect (β)** | **CI-lower** | **CI-upper** | **P** |
| Intercept[#] | 3.929 | 3.608 | 4.250 | <0.001 |
| Protocol B vs A (0 = first; 1 = second) | -0.116 | -0.235 | 0.003 | 0.056 |
| Music Preferred vs Disliked | -0.181 | -0.300 | -0.062 | 0.003 |

[#] Intercept is value measured for reference group, ie. response for preferred music in protocol A.

EPTT: Electrical Pain Tolerance Threshold PPT: Pressure Pain Threshold; CI: confidence interval; mA: Milli-Ampère; N/cm2: Newton (force) per cm2; NRS: Numerical Rating Scale (perceived pain intensity, range 0–10).

## Discussion

The present study investigated the effect of preferred music on pain thresholds in comparison with disliked music in volunteers. The results showed that in both the EPTT and PPT groups mean pain thresholds were significantly higher when volunteers listened to preferred music in comparison with disliked music. Moreover, in both the EPTT and PPT group the overall highest pain thresholds were obtained for preferred music when it was preceded by disliked music. A converse pattern was seen in the outcome of the NRS in both groups, with preferred music corresponding to the lower and disliked music to the higher pain scores, with NRS scores for relative silence interspersed. The differences in the outcomes of this study for perceived pain during listening to preferred versus disliked music were small but this sequence effect "comes along" for listening to disliked music first and secondly to preferred music and "goes off" for listening to preferred music first and secondly to preferred music.

This study needed, according to the power calculation, 170 healthy volunteers per group. Since no previous, comparable studies were available, the number of would-be exclusions was unknown. Therefore, a broad range of volunteers was considered and everyone who met our criteria was included during festival hours. 138 volunteers were included in the EPTT group and 279 volunteers were included in the PPT group. We found that listening to preferred music as well as disliked music both leads to higher pain thresholds. At this moment, is not fully elucidated how music influences pain but distraction has long been suspected to be the main mechanism of action [17]. It has been suggested that the release of endogenous opioids within the brain and spinal cord, in response to pleasurable music, is responsible for its beneficial effects [29,30]. From previous pain research it is known that emotions, elicited by external stimuli, such as films or pictures, are able to modulate pain [31,32]. A sad mood results in higher pain and higher ratings of unpleasantness of painful thermal grill illusions compared to neutral mood states [33]. It has been suggested that basic emotions in music can be identified by its audience and induce corresponding mood states [34]. Therefore it is assumed that music may act accordingly, i.e. by inducing strong positive emotions, resulting in pain relief [35]. This pain reducing effect of music might be attributed to the influence of music on the oxytocin- and opioid-mechanisms in the brain [36,37]. A number of studies have shown that enjoyable or lively music reduced the pain sensation in healthy volunteers, whilst somber and sad music showed an increase in the pain sensation [35,38]. Although not formally tested, our results might indicate that distraction as the underlying mechanism of pain alleviation in this study is unlikely. This because the data showed us that the mean pain thresholds during listening to disliked music were lower than the pain thresholds when listening to no music at all. The endogenous opioid system seems more suited to explain our results. In the literature various types of music were described, mainly chosen by the researchers. It is suggested that musical characteristics such as tempo, harmony, melody and volume might play a role. Based on a systematic review with meta-analyses [39] it could be concluded that instrumental music without lyrics is more effective in pain management of patients. However, bases on this study it was not possible to identify the ideal music characteristics to be used in future pain management. Previous research has shown that listening to well-liked music leads to opioid signalling in certain parts of the brain and the descending pain modulatory system [40]. In addition, preferred music prompts the release of dopamine and opioids in the limbic system, which is thought to be involved in emotion and reward [40]. As a conclusion in the study of Hsieh et al [41] it was stated that well-loved music robustly relieves pain, even over enhanced expectations. As was mentioned before, musically-induced, positive emotions are believed to play a role in the modulation of pain [35]. This might explain the differences in pain thresholds found in this study when preferred music was compared with disliked music. Since volunteers

were not blinded, it is also possible that in the subgroups wherein preferred music was preceded by disliked music, preferred music was more highly anticipated by volunteers and therefore resulted in a stronger sense of reward and subsequent higher pain thresholds. These findings would be consistent with previous research, which stated that the anticipation of an abstract pleasure, e.g. preferred music, leads to dopamine release within the dorsal striatum, which is a part of the reward system [42].

Whilst interpreting the results of this study, several strengths and shortcomings arose which required consideration. Among the strong points were the large study population, the standardised method of testing and the fact that volunteers were able to choose their own preferred and disliked music. Also, the blood alcohol content (BAC) of all volunteers was determined, followed by exclusion in case the BAC exceeded 220 μg/l. Unfortunately, it was neither feasible nor affordable to objectively determine whether volunteers were under the influence of recreational drugs. Instead, we relied on subjective questionnaires and, in questionable cases, the objective judgment of a medical doctor. In this study we specifically chose two different stimuli for the assessment of the individual pain thresholds: PPT on a pain threshold level and EPTT on a pain tolerance level to obtain different information about nociceptive processing in the healthy volunteers. A limitation in this study was that the measurements were only counterbalanced between preferred and disliked music. The silence condition was always assessed first. This might have resulted in an over- or underestimation of reality, due to habituation and sensitisation respectively. The healthy volunteers only received one kind of stimuli because of the availability of measurement equipment. This makes it impossible to compare the two different types of stimuli within the healthy volunteers. In addition, future research should include a pre- and post-test measurement of the pain threshold in order to determine the degree of habituation in volunteers. Moreover, the selected music, preferred as well as disliked, should be noted because musical characteristics might play a role and explain effects beyond preference [39]. Another limitation was that the study population was not a realistic reflection of the average Dutch population, which, as it consisted mostly of younger adults, reduced its generalisability. Previous research has indicated that age may influence sensitivity to noxious and innocuous stimuli, as PPT decrease and somatosensory thresholds for innocuous stimuli increase with age [43]. Finally, a limitation of this study was that the outcomes in disliked music could be an underestimation of the true pain thresholds, as volunteers, in order to stop having to listen to their strongly disliked music, might prematurely indicate their pain threshold had been reached. Unfortunately, this limitation was inherent to the design of the study. To prevent this from happening in future studies, a design might be chosen in which the volunteers have to listen to their chosen piece of disliked music integrally. Another suggestion for future research would be to carry it out in a different study population, e.g. an elderly population with chronic pain or patients undergoing a painful procedure, as this could enhance the external validity and practical implications.

In future studies, the differences between silence, preferred music and disliked music should be further evaluated for their influence on pain perception. By example, the frequencies in the music might play a role in reducing the pain intensity. Besides the effect on pain intensity, there should also be attention for the effect of music on other biopsychochosocial mechanisms, such as fear, anxiety, distress and social interaction.

In conclusion, the present study shows that listening to preferred music when receiving noxious stimuli leads to higher pain thresholds and lower perceived pain scores in comparison with disliked music. Moreover, the highest pain thresholds for both electrical and pressure pain stimuli are obtained when preferred music is preceded by disliked music. Therefore, it is plausible that preferred music could be of benefit when undergoing an actual, painful procedure, especially when it is shortly preceded by disliked music.

## Acknowledgments

We would like to thank all volunteers for participation in this study.

## Author Contributions

**Conceptualization:** Hans Timmerman, Selina E. I. van der Wal, Monique A. H. Steegers.

**Data curation:** Hans Timmerman, Ewald M. Bronkhorst, Monique A. H. Steegers.

**Formal analysis:** Hans Timmerman, Regina L. M. van Boekel, Ludo S. van de Linde, Ewald M. Bronkhorst.

**Funding acquisition:** Hans Timmerman, Kris C. P. Vissers, Monique A. H. Steegers.

**Investigation:** Hans Timmerman, Regina L. M. van Boekel.

**Methodology:** Hans Timmerman, Regina L. M. van Boekel, Selina E. I. van der Wal, Monique A. H. Steegers.

**Project administration:** Hans Timmerman, Monique A. H. Steegers.

**Resources:** Kris C. P. Vissers, Monique A. H. Steegers.

**Supervision:** Hans Timmerman, Kris C. P. Vissers, Monique A. H. Steegers.

**Validation:** Hans Timmerman, Regina L. M. van Boekel, Ludo S. van de Linde, Ewald M. Bronkhorst, Monique A. H. Steegers.

**Visualization:** Hans Timmerman, Regina L. M. van Boekel, Ewald M. Bronkhorst, Monique A. H. Steegers.

**Writing – original draft:** Hans Timmerman, Regina L. M. van Boekel, Ludo S. van de Linde.

**Writing – review & editing:** Hans Timmerman, Regina L. M. van Boekel, Ludo S. van de Linde, Ewald M. Bronkhorst, Kris C. P. Vissers, Selina E. I. van der Wal, Monique A. H. Steegers.

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
