## [Decision Letter · Decision Letter 0]

15 Mar 2022

PONE-D-21-37630The effect of preferred music versus disliked music on pain thresholds in healthy volunteers. A randomized observational study.PLOS ONE

Dear Dr. Timmerman,

Thank you for submitting your manuscript to PLOS ONE. After careful consideration, we feel that it has merit but does not fully meet PLOS ONE’s publication criteria as it currently stands. Therefore, we invite you to submit a revised version of the manuscript that addresses the points raised during the review process.

We look forward to receiving your revised manuscript.

Kind regards,

Lorenzo D. Stafford

Academic Editor

PLOS ONE

Journal Requirements:

Additional Editor Comments (if provided):

The authors have completed a novel study examining the effects of music on objective and subjective pain threshold. I have a few recommendations for the authors to make the article stronger and clearer for readers:

1.L51-"When listening to preferred music compared to relaxation music painful stimuli were tolerated for a longer period, but only in females the experienced pain intensity was lower". Can the authors reflect on why this gender effect and whether it links to other work.

2. L57-"... it is not clear how music influences pain". Some reflection on the possible mechanisms needed here.

3. L77-"during “Lowlands”, a festival located in Biddinghuizen...with a special venue for science and research." Was this a mainly musical festival? Was there any special reason why this specific festival was chosen?

4. L177-I am not clear why the authors analyzed the data using Regression analyses when the data lend themselves much more appropriately to ANOVA. The data should be re-analysed using separate ANOVAs for EPTT and PPT and the main effects for Music Type and Timing and any interactions shown clearly. Additionally, rather than using a composite score (disliked less preferred), the authors should use the existing data for the 3 conditions (silence, disliked, preferred), with the posthoc comparisons clearly showing any differences between the 3 groups.

Reviewers' comments:

Reviewer's Responses to Questions

**Comments to the Author**

1. Is the manuscript technically sound, and do the data support the conclusions?

Reviewer #1: Yes

Reviewer #2: Partly

2. Has the statistical analysis been performed appropriately and rigorously? 

Reviewer #1: Yes

Reviewer #2: I Don't Know

3. Have the authors made all data underlying the findings in their manuscript fully available?

Reviewer #1: Yes

Reviewer #2: No

4. Is the manuscript presented in an intelligible fashion and written in standard English?

Reviewer #1: Yes

Reviewer #2: No

5. Review Comments to the Author

Reviewer #1: Author

Thanks a lot for the opportunity to review this interesting paper. There are a few sentences/concepts that could be communicated more clearly to the reader.

MY COMMENTS ARE AS FOLLOWS

- What was the purpose of randomising the mode of pain stimulus? Please explain your rationale in the text

- L 11 and 12 please rephrase the sentence to make clear that you are randomising the mode of pain stimulus

- L30 and following – the introduction deals with chronic pain. You are testing acute pain in your study protocol, please limit your background/introduction to acute pain

- L154, please state whether participants chose a particular song or piece or a whole genre

- Table 3, the labels intercept, round, music preferred (vs disliked) are not very intuitive, could you consider using more descriptive terms?

- L277-278 I am not sure what that sentence means, please could you re-phrase it? (the differences…)

- L285 we found…higher pain thresholds- compared to what?

- L301 higher pain thresholds were obtained in silence than listening to disliked music (this seems to be in contradiction to L285), perhaps clarify what is your comparator in both cases

- L337 .. were only counterbalanced between… could you please re phrase this sentence to make it clearer

- L338 but leaving out the silence condition – what is meant by that?

I wish you good luck with your publication

Reviewer #2: Introduction – good epidemiological overview of chronic pain. Could expand more on the metanalysis on postoperative pain e.g., more specific details on the main outcome measures. It says pain was an outcome but is this referring to pain intensity, tolerance, interference etc. Line 50 could be made clearer – its unclear what is meant by ‘similar pain tolerance thresholds’. When referring to research findings more generally, it would be good to consistently be specific on the kind of pain outcome that is being referred to. For example, there are a number of instances that state that pain was reduced and it would be helpful to specify the pain outcomes. More details on the music interventions could be included e.g., how long and when would participants be listening to music? How was the music selected? This would provide more context and background information on the use of music as a pain management approach. If possible, it would be good to discuss potential mechanisms that might explain the effect of music on pain e.g., effect on mood or distraction. The use of healthy volunteers in the study could be more strongly justified given the focus on chronic pain.

Method – Line 76, it’s unclear how this study can be both randomised and observational at the same time. Line 84, what was considered ‘normal health’? while it is specified where the study took place, more details could be included in terms of how participants were recruited. For example, where they passing by the venue (opportunistic)? Line 102, needs to be kept in past tense (‘were based…’). It would be good to explain why participants were randomised to either the EPTT or the PPT condition and not be tested using both methods. Line 103 – the acronym ‘QST’ needs to be defined. Line 145-146 – it would be good to state how many operators were physicians, researchers, and students. The start of the ‘study procedures’ section is a bit difficult to follow. It’s unclear how long volunteers listened to music for. It’s stated that they listened to 60 seconds of music before being administered a pain stimulus – did the music stop after they reached their maximum pain tolerance? Line 166 – some grammatical issues here with wording i.e., ‘changed to preferred of disliked music…’

Results – Some incorrect grammar/wording has been used: ‘declination’ and ‘consummation’. These need to be reworded. Details of what ‘protocol A’ vs ‘Protocol B’ involved need to be made clearer in the Methods section. Make sure the word ‘table’ in the main body is always capitalised. The acronyms (PPT and EPTT) should ideally also be used in the main text of the Results section rather than ‘electrical pain thresholds’. When reporting the significance of the findings, it would also be helpful to report effect sizes as an indication of the magnitude of the differences found. The Method section states that regression analyses would be used, but key elements are not reported in the Results section e.g., R, R2, B coefficients.

Discussion – There was some interesting discussion on the mechanisms behind how music may influence pain perception. There were also some good reflections on the strengths and limitations. There could be more stronger discussion on key/wider implications of the study findings. This can include a more critical discussion on future directions and how these findings have contributed to our understanding of pain perception.

Writing – make sure spelling is consistent i.e., ‘randomised’ vs ‘randomized’. While the writing is generally clear, it may be good to seek additional editorial help to check grammar and terminology used.

6. PLOS authors have the option to publish the peer review history of their article (what does this mean?). If published, this will include your full peer review and any attached files.

Reviewer #1: **Yes: **Elizabeth Ball

Reviewer #2: **Yes: **Miznah Al-Abbadey

---

## [Author Response · Author response to Decision Letter 0]

14 Sep 2022

PONE-D-21-37630

The effect of preferred music versus disliked music on pain thresholds in healthy volunteers. An observational study.

Dear Editor,

Thank you for the invitation to submit a revised version of our manuscript titled “The effect of preferred music versus disliked music on pain thresholds in healthy volunteers. A randomized observational study”.

We have included the following items:

• A rebuttal letter that responds to each point raised by the academic editor and reviewer(s). 

• A marked-up copy of our manuscript that highlights changes made to the original version. However, we have left out the changes which were necessary due to the PLOS formatting guidelines. This because of readability.

• An unmarked version of our revised paper without tracked changes. 

• We have uploaded our figure (Figure 1) to the PACE digital diagnostic tool to check and ensure we meet PLOS Requirements.

• In the manuscript it is specified where the minimal data set (including the syntax) underlying the results described in our manuscript can be found. 

Below you will find our answers to the raised issues and questions.

Journal Requirements:

1. Please ensure that your manuscript meets PLOS ONE’s style requirements, including those for file naming. 

Answer: We checked this and adjusted it when necessary. However, we did not use track changes for style requirements because of the readability of the manuscript. For all other revisions in the manuscript we have used track changes. 

2. Please ensure that you include a title page within your main document. 

Answer: We added the title page as part of the manuscript and we have followed the PLOS guideline.

3. Data Availability statement, you have not specified where the minimal data set underlying the results described in your manuscript can be found. 

Answer: We added this information to the manuscript (Page 11, line 251-253)

Answer: We have updated this information in the manuscript (Page 11, line 251-253) as well as in the cover letter. 

The underlying dataset of this study and syntax is available via : Timmerman, H., Boekel, R.L.M. van, Linde, L.S. van de, Bronkhorst, E.M., Vissers, K.C.P., Wal, S.E. van der & Steegers, M.A.H. (2022). The effect of preferred music versus disliked music on pain thresholds in healthy volunteers. An observational study. DANS EASY [Dataset]. DOI: xxx.xxx.xx

The database and syntax are submitted to the DANS EASY repository of the Radboud University Nijmegen, the Netherlands. They will get us a DOI as soon as possible,

Additional Editor Comments:

The authors have completed a novel study examining the effects of music on objective and subjective pain threshold. I have a few recommendations for the authors to make the article stronger and clearer for readers:

1.L51-"When listening to preferred music compared to relaxation music painful stimuli were tolerated for a longer period, but only in females the experienced pain intensity was lower". Can the authors reflect on why this gender effect and whether it links to other work.

Answer: Thanks for the suggestion. We have added this information to the manuscript (Page 4, Line 88-93)

2. L57-"... it is not clear how music influences pain". Some reflection on the possible mechanisms needed here.

Answer: We have added this information to the manuscript (Page 4, Line 94-101)

3. L77-"during “Lowlands”, a festival located in Biddinghuizen...with a special venue for science and research." Was this a mainly musical festival? Was there any special reason why this specific festival was chosen?

Answer: We have adapted this paragraph to make this more clear: “This observational study was conducted in 2016 during “Lowlands”, a festival mainly focussing on music, but also with attention to, by example, literature, ballet and science. This festival is located in Biddinghuizen (the Netherlands) and the organization provides a designated venue for science and research.” (page 6, line 125-128)

4. L177-I am not clear why the authors analyzed the data using Regression analyses when the data lend themselves much more appropriately to ANOVA. The data should be re-analysed using separate ANOVAs for EPTT and PPT and the main effects for Music Type and Timing and any interactions shown clearly. Additionally, rather than using a composite score (disliked less preferred), the authors should use the existing data for the 3 conditions (silence, disliked, preferred), with the posthoc comparisons clearly showing any differences between the 3 groups.

Answer: ANOVA and regression are strongly related techniques. Both look at continuous outcomes but traditionally ANOVA is based on categorical independent variables, while regression analysis uses continuous variables as independent variables. By using dummy variables, categorical independent variables can be coded for use in regression analysis as well. So one might say that regression analysis can do all that ANOVA can do and more. This is correct, but both approaches have extensions. ANCOVA facilitates the inclusion of continuous variables as independent variables in ANOVA analysis. For simple data structures this makes regression and ANCOVA equivalent. However, for data with dependencies such as repeated measurements or data with clustering on one or more levels, the multilevel regression techniques that have been developed to deal with these situations are much further developed than the techniques for the ANOVA bases models. So in short, for relatively simple data structures both approaches are equivalent, while in the more complicated cases regression is further developed and hence more flexible and more effective. In case of this study, ANOVA can easily be used and yields identical estimates of the effects. However, it ignores the repetition of measurements in single participants. And therefore yields estimates of variance that are incorrect. So for this study analysis cannot be improved by changing to an ANOVA based approach.

Reviewer #1: Author

Thanks a lot for the opportunity to review this interesting paper. There are a few sentences/concepts that could be communicated more clearly to the reader.

MY COMMENTS ARE AS FOLLOWS

- What was the purpose of randomising the mode of pain stimulus? Please explain your rationale in the tekst

Answer: Sorry, we were unclear about this. We did not randomize the pain stimulus but we randomly assigned the patient to one of the two stimuli due to the availability of the instruments. We’ve added some extra lines to the manuscript to make this more clear (Page 7, Line 159-166)

- L 11 and 12 please rephrase the sentence to make clear that you are randomising the mode of pain stimulus

Answer: we changed this sentence in “The objective of this observational study was to evaluate the effect of preferred music as opposed to disliked music on pain (tolerance) thresholds and perceived pain intensity in healthy volunteers”. We have addressed the random assignment of the mode of pain stimuli as “The volunteers were randomly assigned to either handheld pressure algometry to assess the pressure pain threshold to or electrical measurements to assess the electrical pain tolerance threshold while listening to preferred and disliked music”. (Page 2, line 27-30)

- L30 and following – the introduction deals with chronic pain. You are testing acute pain in your study protocol, please limit your background/introduction to acute pain

Answer: We have adapted the introduction following your suggestions. However, due to possible transition of acute pain into chronic pain, we think that also the possible influence on chronic pain needs some attention. 

- L154, please state whether participants chose a particular song or piece or a whole genre

Answer: We have added this to the manuscript: “The volunteers were asked to state their preferred song and their most disliked song before commencement of the measurements.” Page 8, line 171-172

“Subsequently, volunteers listened to 60 seconds of their preferred or their disliked music. Afterwards they received pressure or electrical pain stimuli, while they continued listening to their music.” Page 8, line 181-183

- Table 3, the labels intercept, round, music preferred (vs disliked) are not very intuitive, could you consider using more descriptive terms?

Answer: Thanks for the suggestion. We added some more descriptive terms to Table 3. 

- L277-278 I am not sure what that sentence means, please could you re-phrase it? (the differences…)

Answer: We changed the sentences to make this more clear: “The differences in the outcomes of this study for perceived pain during listening to preferred versus disliked music were small but this sequence effect "comes along" for the correct order (listening to disliked music first and secondly to preferred music) and "goes off" for the wrong order (listening to preferred music first and secondly to preferred music. (Page 18, Line 386-390) 

- L285 we found…higher pain thresholds- compared to what?

Answer: we have changed this sentence into: “We found that listening to preferred music as well as disliked music both leads to higher pain thresholds. “Page 18, Line 403-406.

- L301 higher pain thresholds were obtained in silence than listening to disliked music (this seems to be in contradiction to L285), perhaps clarify what is your comparator in both cases

Answer: We made this more clear to the reader: “Although not formally tested, our results might indicate that distraction as the underlying mechanism of pain alleviation in this study is unlikely. This because the data showed us that the mean pain thresholds during listening to disliked music were lower than the pain thresholds when listening to no music at all.” Page 19, Line 412-416

- L337 .. were only counterbalanced between… could you please re phrase this sentence to make it clearer

Answer: We changed this sentence as follows: “A limitation in this study was that the measurements were only counterbalanced between preferred and disliked music. The silence condition was always assessed first. “ (Page 20, Line 448-450)

- L338 but leaving out the silence condition – what is meant by that?

Answer: The assessment of the thresholds without music was always assessed first. The next measurements (with the preferred and disliked music) were counterbalanced. Page 20, Line 448-450)

Reviewer #2: 

Introduction – good epidemiological overview of chronic pain. 

- Could expand more on the metanalysis on postoperative pain e.g., more specific details on the main outcome measures. It says pain was an outcome but is this referring to pain intensity, tolerance, interference etc. 

Answer: We added this information to the manuscript to the introduction paragraph were necessary

- Line 50 could be made clearer – it’s unclear what is meant by ‘similar pain tolerance thresholds’. 

Answer: We added the following information: ”In a study by Hekmat and Hertal [11] it was showed that preferred music increased the pain tolerance and self-reported pain during an ice-water hand immersion test compared to the same task without music. However, in groups with disliked music compared to no music the participants showed no differences in pain tolerance and self-reported pain.” Page 4, Line 83-86

- When referring to research findings more generally, it would be good to consistently be specific on the kind of pain outcome that is being referred to. For example, there are a number of instances that state that pain was reduced and it would be helpful to specify the pain outcomes. 

Answer: Thanks for the suggestion. We added this information where necessary.

- More details on the music interventions could be included e.g., how long and when would participants be listening to music? How was the music selected? This would provide more context and background information on the use of music as a pain management approach. 

Answer: We have added this information to the methods paragraph. Page X, Line Y-Z

- If possible, it would be good to discuss potential mechanisms that might explain the effect of music on pain e.g., effect on mood or distraction. 

Answer: We have added this information to the introduction paragraph. Page 4, Line 94-101

- The use of healthy volunteers in the study could be more strongly justified given the focus on chronic pain.

Answer: Our focus is more on acute pain than on chronic pain. Following the first reviewer we have partially rewritten our introduction section with more focus on acute pain. We have included healthy volunteers and gave them a short painful stimulation, which is more in line with acute pain. 

Method

- Line 76, it’s unclear how this study can be both randomised and observational at the same time. 

Answer: We are sorry for this confusing statement. As this study has an observational study design, we only randomized the healthy volunteers into the type of stimuli they would receive (pressure pain threshold assessment of electrical pain tolerance threshold) and if they started to listen to preferred music or disliked music. We added this information to the method section, paragraph “study procedures” and deleted the heading randomization to be more clear this is a observational study (Page 6, line 149)

- Line 84, what was considered ‘normal health’? 

Answer: ‘Normal health’ was in this study when there was no reported disease and/or illness. We added this also to the manuscript: “ (2) in normal health (no reported diseases and/or illness)”. (Page 6, line 137)

- While it is specified where the study took place, more details could be included in terms of how participants were recruited. For example, where they passing by the venue (opportunistic)? 

Answer: We added this to the method section: “Interested volunteers were asked to participate by the researchers of this study when they were passing by the research venue. When they were (1) at least 18 years old, (2) in normal health (no reported diseases and/or illness) and (3) had sufficient knowledge of the Dutch language they were able to participate in this study. (Page 6, line 134-136) 

- Line 102, needs to be kept in past tense (‘were based…’). 

Answer: Our mistake, we have adapted it following your suggestion.

- It would be good to explain why participants were randomised to either the EPTT or the PPT condition and not be tested using both methods. 

Answer: This was because of the availability of the measurement equipment in the measurement stations (two boots for electric pain thresholds and four boots for pressure pain thresholds). During the festival volunteers were visiting the location. To keep up a good flow of the participants we choose not to change boots but only to perform one type of measurement instead of two. We added this also to the discussion section: (Page 21, 456-458). “The healthy volunteers only received one type of stimuli because of the availability of measurement equipment. This makes it impossible to compare the two different types of stimuli within the healthy volunteers.”

-Line 103 – the acronym ‘QST’ needs to be defined. 

Answer: we changed this line in: “The measurements obtained in this study were based on the methodology of the Nijmegen Aalborg Screening Quantitative Sensory Testing (QST), also called NASQ protocol.[17-20]” (Page 7, line 150-152)

-Line 145-146 – it would be good to state how many operators were physicians, researchers, and students. 

Answer: We added this to the methods section, study procedures paragraph: “The measurements during the festival were performed by four physicians, four researchers and four students.” (Page 7, 152-153).

-The start of the ‘study procedures’ section is a bit difficult to follow. It’s unclear how long volunteers listened to music for. It’s stated that they listened to 60 seconds of music before being administered a pain stimulus – did the music stop after they reached their maximum pain tolerance? 

Answer: We have added this information more clearly to the study procedures. Page 8, Line 181-189

-Line 166 – some grammatical issues here with wording i.e., ‘changed to preferred of disliked music…’

Answer: We have corrected it as: “The music started again with either preferred or disliked music depending on the study protocol (protocol A or B) and the procedure and measurements were repeated.” Page 8, Line 187-189.

Results – Some incorrect grammar/wording has been used: ‘declination’ and ‘consummation’. These need to be reworded. 

Answer: Thanks, we re-read the manuscript carefully on these kind of mistakes.

-Details of what ‘protocol A’ vs ‘Protocol B’ involved need to be made clearer in the Methods section. 

Answer: We have added this to the methods section, page 7, line 163-166

-Make sure the word ‘table’ in the main body is always capitalised. 

Answer: We have corrected this in the manuscript where necessary

-The acronyms (PPT and EPTT) should ideally also be used in the main text of the Results section rather than ‘electrical pain thresholds’. 

Answer: We have corrected this in the manuscript where necessary

-When reporting the significance of the findings, it would also be helpful to report effect sizes as an indication of the magnitude of the differences found. 

Answer: In the text of the manuscript we gave the p-values and the effect.

The Method section states that regression analyses would be used, but key elements are not reported in the Results section e.g., R, R2, B coefficients.

Answer: The B-coefficient is the effect. We added (β) to the effect in Table 3. We did not give R /R2 because they are less interpretable in a mixed model. 

Discussion – There was some interesting discussion on the mechanisms behind how music may influence pain perception. There were also some good reflections on the strengths and limitations. 

-There could be more stronger discussion on key/wider implications of the study findings. This can include a more critical discussion on future directions and how these findings have contributed to our understanding of pain perception.

Answer: We have added this to the discussion section: “In future studies, the differences between silence, preferred music and disliked music should be further evaluated for their influence on pain perception. By example, the frequencies in the music might play a role in reducing the pain intensity. Besides the effect on pain intensity, there should also be attention for the effect of music on other biopsychochosocial mechanisms, such as fear, anxiety, distress and social interaction.” Page 21, Line 473-478

Writing – make sure spelling is consistent i.e., ‘randomised’ vs ‘randomized’. While the writing is generally clear, it may be good to seek additional editorial help to check grammar and terminology used.

Answer: Sorry for our mistakes We checked the spelling and grammar again.

---

## [Decision Letter · Decision Letter 1]

2 Nov 2022

PONE-D-21-37630R1The effect of preferred music versus disliked music on pain thresholds in healthy volunteers. An observational study.PLOS ONE

Dear Dr. Timmerman,

Thank you for submitting your manuscript to PLOS ONE. After careful consideration, we feel that it has merit but does not fully meet PLOS ONE’s publication criteria as it currently stands. Therefore, we invite you to submit a revised version of the manuscript that addresses the points raised during the review process.

We look forward to receiving your revised manuscript.

Kind regards,

Lorenzo D. Stafford

Academic Editor

PLOS ONE

Journal Requirements:

Reviewers' comments:

Reviewer's Responses to Questions

**Comments to the Author**

1. If the authors have adequately addressed your comments raised in a previous round of review and you feel that this manuscript is now acceptable for publication, you may indicate that here to bypass the “Comments to the Author” section, enter your conflict of interest statement in the “Confidential to Editor” section, and submit your "Accept" recommendation.

Reviewer #2: (No Response)

2. Is the manuscript technically sound, and do the data support the conclusions?

Reviewer #2: Yes

3. Has the statistical analysis been performed appropriately and rigorously? 

Reviewer #2: Yes

4. Have the authors made all data underlying the findings in their manuscript fully available?

Reviewer #2: Yes

5. Is the manuscript presented in an intelligible fashion and written in standard English?

Reviewer #2: No

6. Review Comments to the Author

Reviewer #2: In relation to question 5 (writing style), there has been a significant improvement in the writing since the first submission. However, I noticed a few more errors/ambiguities that I have noted below.

Introduction – line 73-76 needs the citation. Line 83, the citation “[14]” would ideally need to be moved to after the author names, rather than at the end of the sentence to be consistent with how this was done on line 78 (“Bradt and colleagues [12]”). The alternative would be to adjust line 78 so that the citation is at the end of the sentence. Line 84 – do the authors mean that preferred music increased pain tolerance and lowered self-reported pain intensity?

Method – line 188, NRS needs to be defined as it is first being used here. It might be good to include the ‘measurements’ section before the ‘Procedure’ section, as this defines all the acronyms and measures.

Results – line 266, use of the term ‘consummated’ is unusual here. This could be changed to ‘consumed’ or ‘ingested’. Lines 284-286, it’s usually good to also include some indication of dispersion e.g., SD or the range alongside the averages. Line 343, there is a spelling error/typo- “response”.

Discussion – line 360, the word “also” can be deleted. Lines 364-368, this sentence is quite wordy and makes an assumption that there is a ‘correct’ or ‘incorrect’ order of music choice. Although listening to disliked music first followed by liked music seemed to lead to better pain outcomes, describing the order of music in this way seems less objective. Lines 378-380, this sentence needs a citation. Line 399, its unclear what is meant by “no optimal characteristics for pain management were given” – it would be good to elaborate on this point more. Line 429, there is repetition of the word “of”.

7. PLOS authors have the option to publish the peer review history of their article (what does this mean?). If published, this will include your full peer review and any attached files.

Reviewer #2: No

---

## [Author Response · Author response to Decision Letter 1]

2 Dec 2022

See also our rebuttal letter for our answers to the highly appreciated comments by the reviewers and the editor.

Below we have copied our answers:

PONE-D-21-37630R1

The effect of preferred music versus disliked music on pain thresholds in healthy volunteers. A randomized observational study.

Dear Editor,

Thank you for the invitation to submit a revised version of our manuscript titled “The effect of preferred music versus disliked music on pain thresholds in healthy volunteers. A randomized observational study”.

We have included the following items:

• A rebuttal letter that responds to each point raised by the academic editor and reviewer(s). 

• A marked-up copy of our manuscript that highlights changes made to the original version. However, we have left out the changes which were necessary due to the PLOS formatting guidelines. This because of readability.

• An unmarked version of our revised paper without tracked changes. 

• We have uploaded our figure (Figure 1) to the PACE digital diagnostic tool to check and ensure we meet PLOS Requirements.

• In the manuscript it is specified where the minimal data set (including the syntax) underlying the results described in our manuscript can be found, with the DOI. 

Below you will find our answers to the raised issues and questions.

Journal Requirements:

We have adapted the reference list to Vancouver style instead of the PLOS style in EndNote. Sorry for this mistake. Moreover, we have checked the reference list on completeness and correctness. No retracted papers were cited. 

Reviewer comments:

Reviewer #2: In relation to question 5 (writing style), there has been a significant improvement in the writing since the first submission. However, I noticed a few more errors/ambiguities that I have noted below.

Dear reviewer, thanks for your comments and suggestions!

Introduction: 

Line 73-76 needs the citation. 

 We added reference [11] to this line. (line 78-80)

Line 83, the citation “[14]” would ideally need to be moved to after the author names, rather than at the end of the sentence to be consistent with how this was done on line 78 (“Bradt and colleagues [12]”). The alternative would be to adjust line 78 so that the citation is at the end of the sentence. 

 We have replaced the references in the manuscript were necessary.

Line 84 – do the authors mean that preferred music increased pain tolerance and lowered self-reported pain intensity?

Our mistake, we have corrected it following your remark. (line 83-86)

Method:

Line 188, NRS needs to be defined as it is first being used here. 

 We have checked the acronym. (line 180)

It might be good to include the ‘measurements’ section before the ‘Procedure’ section, as this defines all the acronyms and measures.

We have placed the measurement section before the procedure section following your suggestion and checked the acronyms throughout the manuscript

Results:

Line 266, use of the term ‘consummated’ is unusual here. This could be changed to ‘consumed’ or ‘ingested’. 

 We have changed ‘consummated’ into ‘consumed’ (line 265)

Lines 284-286, it’s usually good to also include some indication of dispersion e.g., SD or the range alongside the averages. 

 We have added the standard deviations to the stated avarages. (line 282-286)

Line 343, there is a spelling error/typo- “response”.

We have corrected this typo. (line 367-368)

Discussion: 

Line 360, the word “also” can be deleted. 

 We have deleted this word. (line 383)

Lines 364-368, this sentence is quite wordy and makes an assumption that there is a ‘correct’ or ‘incorrect’ order of music choice. Although listening to disliked music first followed by liked music seemed to lead to better pain outcomes, describing the order of music in this way seems less objective. 

 We have corrected this from:

“The differences in the outcomes of this study for perceived pain during listening to preferred versus disliked music were small but this sequence effect "comes along" for the correct order (listening to disliked music first and secondly to preferred music) and "goes off" for the wrong order (listening to preferred music first and secondly to preferred music).” 

Into:

“The differences in the outcomes of this study for perceived pain during listening to preferred versus disliked music were small but this sequence effect "comes along" for listening to disliked music first and secondly to preferred music and "goes off" for listening to preferred music first and secondly to preferred music.” (Line 387-390)

Lines 378-380, this sentence needs a citation. 

We added the required references. (Line 400-402)

Line 399, it’s unclear what is meant by “no optimal characteristics for pain management were given” – it would be good to elaborate on this point more. 

 We agree this was unclear. We have re-written this as:

“It is suggested that musical characteristics such as tempo, harmony, melody and volume might play a role. Based on a systematic review with meta-analyses [38] it could be concluded that instrumental music without lyrics is more effective in pain management of patients. However, based on this study it was not possible to identify the ideal music characteristics to be used in future pain management.” (Line 418-423)

Line 429, there is repetition of the word “of”.

 We have corrected this in the manuscript. (Line 454)

---

## [Editor Report · Decision Letter 2]

21 Dec 2022

The effect of preferred music versus disliked music on pain thresholds in healthy volunteers. An observational study.

PONE-D-21-37630R2

Dear Dr. Timmerman,

We’re pleased to inform you that your manuscript has been judged scientifically suitable for publication and will be formally accepted for publication once it meets all outstanding technical requirements.

Kind regards,

Lorenzo D. Stafford

Academic Editor

PLOS ONE
---

## [Editor Report · Acceptance letter]

4 Jan 2023

PONE-D-21-37630R2 

The effect of preferred music versus disliked music on pain thresholds in healthy volunteers. An observational study 

Dear Dr. Timmerman:

I'm pleased to inform you that your manuscript has been deemed suitable for publication in PLOS ONE. Congratulations! Your manuscript is now with our production department. 

Kind regards, 

on behalf of

Dr. Lorenzo D. Stafford 

Academic Editor

PLOS ONE